# Effects of the γ″-Ni_3_Nb Phase on Mechanical Properties of Inconel 718 Superalloys with Different Heat Treatments

**DOI:** 10.3390/ma13010151

**Published:** 2019-12-31

**Authors:** Li-Shi-Bao Ling, Zheng Yin, Zhi Hu, Jin-Hui Liang, Zhi-Yong Wang, Jun Wang, Bao-De Sun

**Affiliations:** 1Shanghai Key Lab of Advanced High-Temperature Materials and Precision Forming, Shanghai Jiao Tong University, Shanghai 200240, China; zhengyin1114@163.com (Z.Y.); junwang@sjtu.edu.cn (J.W.); bdsun@sjtu.edu.cn (B.-D.S.); 2Mechanical and Electrical Engineering College, Nanchang University, Nanchang 330031, China; huzhi@ncu.edu.cn; 3Beijing Institute of Near Space Vehicle System Engineering, Beijing 10000, China; I.night@163.com; 4AVIC Guizhou Anji Aviation Precision Casting Co., Ltd., Guizhou 561000, China; GZWZY73@163.com

**Keywords:** γ″-Ni_3_Nb, mechanical properties, Inconel 718 superalloy, heat treatment, hot isostatic pressing

## Abstract

The effects of the γ″-Ni_3_Nb phase on the mechanical properties of Inconel 718 superalloys, with standard heat treatment, hot isostatic pressing + solution treatment + aging, and hot isostatic pressing + direct aging, were characterized by morphological observation, X-ray diffraction, and tensile experiments. The results of the morphological observation revealed that many fine γ″ precipitates of ~26.49 ± 1.82 nm in mean size were formed in all samples. However, the relatively coarser γ″ precipitates formed in the grain boundaries were only observed in the sample treated with hot isostatic pressing + direct aging. The yield strengths of the hot isostatic pressing + direct aging sample at room temperature and at 650 °C both exhibited the maximum values about 993 ± 5.7 and 811 ± 12.6 MPa, respectively. The γ″ precipitate was considered to be the dominant strengthening phase in the sample according to the lattice misfits (ε) of γ/γ″. The strengthening mechanism of the samples can be explained as the coherency strain strengthening of fine γ″ precipitates. Moreover, due to the coarser γ″ precipitates in the grain boundaries, dislocation-cut ordered particle strengthening also occurred in the sample after hot isostatic pressing + direct aging treatment.

## 1. Introduction

Inconel 718 superalloys have been used for a wide range of applications because of their excellent qualities such as good comprehensive mechanical properties, easy processing, relatively low cost, and outstanding weldability [1,2,3,4,5,6]. These materials are extensively used in aerospace engines, gas turbines, and nuclear and petrochemical industries [7]. The primary casting process for manufacturing complex superalloy components is investment casting [8,9]. However, the related solidification defects, such as microporosity, dendritic segregation, and the formation of Laves and δ phases, have harmful effects on the mechanical properties of the cast superalloy [10]. To address this problem, hot isostatic pressing is required to close pores, homogenize the microstructure, and improve the mechanical properties of the cast Inconel 718 superalloy.

In recent years, increasing attention has been focused on the hot isostatic pressing (HIP) of the Inconel 718 alloy which has a fine uniform microstructure and good mechanical properties [11,12,13,14]. Chang et al. [15] indicated that HIP treatment of the Inconel 718 superalloy can make the grain uniform, improve the segregated structure, and decrease the porosity by 85.8%. Under the conditions of the most appropriate temperature (i.e., 1230 °C) and pressure (i.e., 120 MPa) for 4 h, the creep strength, fatigue strength, and toughness of the GTD-111-DS superalloy were improved by four times [16]. Qiu et al. [17] found that the ultimate tensile strength and elongation increased considerably, but that the 0.2% yield strengths of the nickel-based superalloy at room temperature and 700 °C decreased when the HIP temperature increased. However, extensive research efforts have been focused on verifying the effectiveness of HIP and the optimization of parameters [18,19]. To further improve the mechanical properties, a heat treatment regime (standard heat treatment, solution treatment, and direct aging) is recommended to realize the good strength and ductility of the HIP Inconel 718 superalloy [3,7].

Generally, HIP and heat treatment of the Inconel 718 superalloy are key steps that influence its microstructure and mechanical properties [20]. Heat treatment and the HIP process can influence the microstructural evolution during processing thus improving the mechanical performance of the Inconel 718 superalloy [21,22,23,24]. Liu et al. [25] pointed out that the incorporation of HIP and a standard heat treatment can improve the strength and fatigue life by eliminating the microporosity of nickel-based superalloys. Qin et al. [26] found that the γ″ phase presented homogeneous nucleation, and the decrease in strength of Inconel 718 superalloy was mainly due to the size reduction of the γ″ precipitates. However, the strengthening mechanisms of γ″ precipitates on the mechanical properties of the Inconel 718 superalloys after hot isostatic pressing and post-heat treatment have rarely been reported [27]. Therefore, in the present work, the effects of the morphological evolution of γ″ precipitates on the mechanical properties of the Inconel 718 superalloys after hot isostatic pressing and post-heat treatment were studied.

## 2. Experimental Procedures

### 2.1. Materials Preparation

The material used in this study was Inconel 718 superalloy which had an actual composition, measured by inductively coupled plasma atomic emission spectrometry, of (wt.) 52.1% Ni, 19.5% Cr, 4.8% Nb, 3.3% Mo, 0.6% Ti, 0.4% Al, 0.06% C, 0.19% Si, 0.003% P, and Fe balance. The Inconel 718 superalloy ingot was melted in a vacuum induction furnace to produce investment casting with a ceramic shell preheated at 200 °C. The specimens were then subjected to three treatments: standard heat treatment (SHT), hot isostatic pressing + solution treatment + aging (HIP + STA), and hot isostatic pressing + direct aging (HIP + DA). According to the standard heat treatment (ASTM B637), the SHT was carried out with the regime of 1095 °C × 1 h/air cooling + 955 °C × 1 h/air cooling + 720 °C × 8 h/furnace cooling at 56 °C/h to 620 °C × 8 h/air cooling to homogenize the original morphology of the microstructure and to increase the amount of precipitates. The HIP was performed at a pressure of 130 MPa and a temperature of 1170 °C for 4 h under an argon atmosphere which could eliminate the microporosity. The HIP + STA was a combination of HIP and STA (955 °C × 1 h/air cooling + 720 °C ×8 h/furnace cooling at 56 °C/h to 620 °C × 8 h/air cooling). The HIP + DA was implemented with the regime of HIP + direct aging (720 °C × 8 h/furnace cooling at 56 °C/h to 620 °C × 8 h/air cooling).

### 2.2. Methodology of Metallographic Investigation

For microstructural analysis, all samples were grinded with 400^#^–2000^#^ silicon carbide (SiC) paper and polished with 5–0.5 μm diamond paste on an metallographic polishing machine (MP-2A, Shanghai, China). The surfaces of all samples were chemically etched by a mixed solution of 5 g CuCl_2_ + 100 mL C_2_H_5_OH + 100 mL HCl for several seconds. The samples were then ultrasonically cleaned in ethyl alcohol and then dried in cold air. The microstructures of the samples were observed by an optical microscope (OM, Olympus BX51M, Tokyo, Japan) and scanning electron microscope (SEM, TESCAN, Klíčany, Czech Republic) equipped with energy-dispersive spectroscopy (EDS, OXFORD, London, Britain). The deformation behavior and dislocation–precipitate interactions of the Inconel 718 superalloy were observed via transmission electron microscopy (TEM, JEOL-2100F, Tokyo, Japan) at 200 kV. The samples were mechanically polished into thin foils with a diameter of 3 mm and thicknesses of 70 μm. Before the experiment, the thin foils were placed in an twin-jet electro polishing machine (MTP-1A, Beijing, China) with an electrolyte solution of 5% HClO_4_ and 95% C_2_H_5_OH using a liquid nitrogen cold region to maintain the temperature between −30 and −15 °C. The lattice parameters of different precipitates and the misfit values between the precipitates and matrix in the Inconel 718 superalloys were characterized by slow scan X-ray diffraction (XRD, Bruker, Karlsruhe, Germany) with CuK_α1_ radiation (λ = 0.154056 nm) in the angular range of 20° ≤ 2θ ≤ 100°, ∆2θ = 0.02°.

### 2.3. Tensile Experiments

Figure 1 presents the dimensions of the standard tensile specimens. Gauge sections of the round-bar specimens with diameters of Φ2.5 were used in tensile tests at room temperature and at 650 °C. The tensile tests at room temperature were conducted according to the ISO 6892-1-2009 standard, and the tensile tests at 650 °C were conducted according to the ISO 783-1999 standard. The high-temperature testing was performed under an argon atmosphere with a flow rate of 5–18 L/h. All tensile tests were performed under a constant strain rate of 6.5 × 10^−6^ s^−1^ using a universal tensile testing machine (CMT5105, Zhejiang, China). All measurements of tensile samples were repeated five times to reduce errors. After removing the highest value and the lowest value, the data were recorded. The surface morphologies of the fracture specimens were analyzed by SEM.

## 3. Results

### 3.1. Microstructural Characterization

Figure 2 shows the optical microstructures of the Inconel 718 superalloy after different heat treatments. As shown in Figure 2a, the typical dendrite structures in the cast Inconel 718 superalloy consisted of bright white Laves phases and black “island” segregation regions. Moreover, there existed some grey cubic MC-carbide particles in the matrix. Compared to the as-cast structure, the regions of segregation gradually disappeared in the case of the standard heat treatment (Figure 2b). It can be clearly observed that the Laves phases were surrounded by acicular δ phases at the inter-dendritic and grain boundaries. Figure 2c reveals that long acicular δ precipitates were distributed discretely around the grain boundaries in the HIP + STA sample. The further DA after the HIP process can form many fine network grain boundaries, and there still existed some segregated regions in the HIP + DA sample as shown in Figure 2d. It is worth noting that the MC-carbide particles were present in all the experiment results as shown in Figure 2.

Figure 3 presents the SEM images of the Inconel 718 superalloy after different heat treatments. Table 1 includes the results from the EDX analysis of the Inconel 718 superalloy after different heat treatments in Figure 3. As shown in Figure 3a and Table 1, the SEM micrographs of the SHT sample exhibit equiaxed Laves phases and adjacent long acicular δ phases. It can be seen that the fine disk-like γ″ phases were evenly distributed in the γ-matrix of the amplificatory image (Figure 3b). As displayed in Figure 3c, few gray Laves phases were distributed over the bright white acicular δ (Ni_3_Nb) phases in the HIP + STA sample. Combined with the results from the EDX, point 4 in Figure 3c was identified to be MC-carbide particles. Figure 3d shows many smaller γ″ phases in the matrix and some bright white δ phases in the grain boundaries. After the treatment of HIP + DA (Figure 3e), there were many fine γ″ precipitates in the matrix and coarser γ″ precipitates in the grain boundaries. Moreover, the coarser γ″ precipitates with major axes of ~200 nm and minor axes of ~60 nm exhibited the same phase relationship in the amplificatory image of Figure 3f.

To intuitively identify the differences in the precipitation behavior of γ″ phases, selected area electron diffraction (SAED) patterns and dark field images of γ″ phases in the SHT, HIP + STA, and HIP + DA Inconel 718 superalloys were obtained and are presented in Figure 4. The analysis of the SAED pattern reveals that the γ″ phase was characterized by an electron beam parallel to the [001] zone in all specimens (Figure 4a). As revealed by dark field images in Figure 4b–d, the γ′ and γ″ phases precipitated in the grains in a cross array in which the γ″ phases were disc-like miniature particles in all samples. The volume fractions and mean sizes of the γ″ precipitates in the SHT, HIP + STA, and HIP + DA Inconel 718 superalloys were measured using an Image-Pro Plus 6.0 image analyzer (Media Cybernetics, Rockville, MD, USA) (Figure 5). The width and length of the images being measured were 606 and 607 pixels. All measurements were repeated five times to reduce errors. Figure 5 shows that the volume fraction of γ″ precipitates was significantly higher in the HIP + STA sample (34.43% ± 1.73%) than in the SHT (17.18% ± 0.56%) and HIP + DA (20.71% ± 0.93%) samples. Further, the mean size of γ″ precipitates in the HIP + STA sample was the finest (16.37 ± 0.93 nm) and that of the HIP + DA sample was the largest (26.49 ± 1.82 nm); these findings are consistent with the TEM observations.

### 3.2. Lattice Misfit

Figure 6 presents the respective slow scan X-ray diffractions and corresponding deconvolutions of diffraction peak 3 of the SHT, HIP + STA, and HIP + DA Inconel 718 superalloys. As shown in Figure 6a, all specimens contained γ matrix, γ′-Ni_3_[Al, Ti], γ″-Ni_3_Nb phase, and a small amount of [Nb, Ti]C particles. However, there was a small amount of diffraction peaks of Laves phase (Fe_2_Nb) and few δ (Ni_3_Nb) phase in the SHT Inconel 718 superalloy. Each of the diffraction peaks was actually an overlap of peaks associated with the individual phases, because the lattice constants of γ, γ′, and γ″ phases were very close [28]. The diffraction peaks of the γ, γ′, and γ″ phases were obtained by Origin PFM software. To accurately determine the lattice parameters of the different precipitates and the lattice misfit values between the precipitates and the matrix, the diffraction peaks 3 of SHT, HIP + STA, and HIP + DA Inconel 718 superalloys are shown in Figure 6b–d in an expanded diagram on the 2θ axis. In the peak fitting results, three peaks, namely, γ″ (220), γ′ (022), and γ (022), at the 2θ axis at about 74.5° can be fitted to the data. The lattice parameters and lattice misfits of the three precipitates in all samples were calculated using an extrapolating function method based on the Bragg equation [29] and relevant crystal structure, and the calculated values are listed in Table 1.

As revealed in Table 2, the lattice constants of the γ′ and γ″ phases for the HIP + DA Inconel 718 superalloy were larger than those of the others which suggest that there is a large lattice distortion. Accordingly, the lattice misfits (ε) of γ/γ′ and γ/γ″ in the HIP + DA Inconel superalloy had the largest values of 0.1928% and 0.6320%, respectively. It is also worth noting that the lattice misfit of γ/γ′ was smaller than that of γ/γ″ in all specimens which indicates that the strengthening γ′ precipitates had a smaller effect in improving the mechanical properties than γ″ precipitates in the Inconel 718 superalloys.

### 3.3. Mechanical Properties

Figure 7 shows the tensile properties of the SHT, HIP + STA, and HIP + DA Inconel 718 superalloys at room temperature and at 650 °C. Table 2 lists the mean values of the yield strength (YS), ultimate tensile strength (UTS), and elongation (EL) of the SHT, HIP + STA, and HIP + DA samples at room temperature and at 650 °C. Compared with the SHT sample, the HIP+STA and HIP + DA samples had a significant improvement in 0.2% YS (increases of ~158 MPa and ~179 MPa, respectively), UTS (increases of ~78 MPa and ~118 MPa, respectively,) and EL (increases of ~1.2% and ~2.8%, respectively) at room temperature. Furthermore, the HIP + DA Inconel 718 superalloy possessed the best mechanical properties at room temperature. For the 650 °C tensile experiment, the HIP + STA and HIP + DA Inconel 718 superalloys presented better strength and ductility compared to the SHT sample. As listed in Table 3, the increase in testing temperature (from room temperature to 650 °C) caused decreases of ~171 MPa and ~225 MPa in both the 0.2% yield strength and ultimate tensile strength, respectively, but only caused a limited change in elongation. For the purpose of comparison, the AMS 5662G [30] and 5596 [31] specification values for wrought (forged and heat treated) material are also included in this Table 3. The values of strength obtained in this study were still lower than those of materials produced by casting and forging, and the reason for this could be because of material that did not have full density. Very lower elongation corresponding, especially unit length, supported this idea.

Figure 8 displays the SEM fractographs of the tensile tested specimens of the SHT, HIP + STA, and HIP + DA Inconel 718 superalloys at room temperature and at 650 °C. As shown in Figure 8a,d, the fractographs of the SHT specimens at room temperature and at 650 °C present many cleavage striations, a few dimples, and micro-holes. After HIP + STA treatment, the fractographs at room temperature were mainly composed of cleavage striations and many small dimples (Figure 8b), while those at 650 °C were mainly composed of many dimples with different sizes and shallow depths (Figure 8e). Therefore, it can be inferred that the fracture modes at room temperature are intergranular and transgranular fractures, while the fracture mode at 650 °C is transgranular ductile fracture. After HIP + DA treatment, the fractographs both at room temperature and 650 °C showed mostly large and deep dimples with equiaxial and parabolic shapes which was related to the elimination of the acicular δ phase as shown in Figure 8c,f.

## 4. Discussions

The strengthening phases of the Inconel 718 superalloy were the body-centered tetragonal γ″ precipitates and face-centered cubic γ′ precipitates. The γ″ precipitate is the dominant strengthening phase due to the lattice misfit (ε) of γ/γ″. The strengthening mechanism of γ″ precipitates can be classified as a coherent strain strengthening mechanism and a dislocation-cut ordered particles mechanism which are related to the content, size, and distribution of γ″ precipitates [32,33]. According to the coherent strain strengthening mechanism, the decisive role in the strengthening of Inconel 718 superalloy is the interaction of the motion dislocations and the stress field generated around the γ″ precipitates. The increase in yield strength ($\Delta \tau_{1}$) caused by coherent strain strengthening during critical resolved shear stress can be calculated by the follow equation [34,35]:(1)$$\Delta \tau_{1}=1.7\mu {\left|\varepsilon_{\gamma \prime \prime }\right|}^{3/2}{\left\{\frac{f_{\gamma \prime \prime }\left(1-\beta \right)}{2b}\right\}}^{1/2}\frac{h_{1}}{R_{1}^{1/2}}$$
where *μ* (72.56 GPa) is the shear modulus of the matrix, $\varepsilon_{\gamma \prime \prime }$ is the lattice misfit between the γ″ precipitate and the matrix, $f_{\gamma \prime \prime }$ is the volume fraction of the γ″ precipitate, *b* (0.2563 nm) is the Burgers vector of the glide dislocation, and *β* (1/3) is the constant related to the distribution of γ″ precipitates. $\frac{h_{1}}{R_{1}^{1/2}}$ is the shape factor of the γ″ precipitates, where *R*_1_ and *h*_1_ represent the semi-major and semi-minor axes lengths of the γ″ precipitates.

The calculated results ($\Delta \tau_{1}$) of the SHT, HIP + STA, and HIP + DA Inconel 718 superalloys are listed in Table 4. It can be seen that the values of yield strength for the HIP + STA (337.04 MPa) and HIP + DA (316.43 MPa) Inconel 718 superalloys were higher than that of the SHT Inconel 718 superalloy. However, the theoretically calculated value of the HIP + DA alloy had a lower yield strength increase than the HIP + STA alloy which is not in agreement with the experimental value measured by the tensile test. Due to the coarser γ″ precipitates in the grain boundaries of the HIP + DA Inconel 718 superalloy, the yield behavior and yield strength were not only affected by the coherent strain strengthening mechanism but also by another strengthening mechanism.

The interaction of dislocations with γ″ precipitates will become complicated and diverse due to the network distribution of the coarser γ″ precipitates in the HIP + DA Inconel 718 superalloy. Precipitated particles can impede the motion of dislocations through a variety of interaction mechanisms. The interaction behavior of the precipitated particles with dislocations is significant for improving the strength. Figure 9 presents the TEM micrographs of the deformed region of the tensile specimen after HIP + DA treatment. As shown in Figure 9a, the coarser γ″ precipitates were repeatedly cut (indicated by an arrow) by dislocation during the tensile deformation process. In addition, there were many high-density paired motion dislocations in another deformed region (Figure 9b). Therefore, it can be determined that the strengthening mechanism caused by the hindrance of the relative dislocation of the coarser γ″ precipitates was the dislocation-cut ordered particles mechanism. The increased value in yield strength ($\Delta \tau_{2}$) caused by dislocation-cut ordered particle strengthening can be calculated by the equation [36,37]:(2)$$\Delta \tau_{2}=\frac{\gamma_{APB^{\prime \prime }}}{2b}\left\{\left[\frac{4\gamma_{APB\prime \prime }f_{\gamma \prime \prime -l}}{\pi T}{\left(\frac{\surd 6R_{2}h_{2}}{3}\right)}^{1/2}\right]-\beta f_{\gamma \prime \prime -l}\right\}$$
where $\gamma_{APB\prime \prime }$ (298 mJ/m^2^) is the antiphase boundary energy, $f_{\gamma \prime \prime -l}$ is the volume fraction of coarser γ″ precipitates, which is 4.8% in the HIP + DA Inconel 718 superalloy, *R* (85 nm) and *h* (13 nm) are the semi-major and semi-minor axes lengths of the coarser γ″ precipitates, and $T=\mu b^{2}/2$; β (1/3) is the constant. As listed in Table 3, the increase in the yield strength of the HIP + DA Inconel 718 superalloy (359.62 MPa) was due to the value of coherent strain strengthening (316.43 MPa) and the value of the dislocation-cut ordered particle strengthening (43.19 MPa) and is consistent with the experimental value. Therefore, the higher strength of the HIP + DA Inconel 718 superalloy was the result of the coherent strain strengthening of the matrix nanoscale γ″ phase and the dislocation cut through the coarser γ″ phase.

## 5. Conclusions

The effects of the γ″ phase on mechanical properties of Inconel 718 superalloys with treatments of SHT, HIP + STA, and HIP + DA were investigated in this study. Based on the present research, several conclusions can be summarized as follows:The relatively coarser γ″ precipitates formed in the grain boundaries were only observed in the superalloy treated with HIP + DA. Moreover, Inconel 718 superalloys with heat treatments exhibited many fine γ″ precipitates of ~26.49 ± 1.82 nm in mean size.The yield strengths of the HIP + DA Inconel 718 superalloy at room temperature and at 650 °C both possessed the maximum values which were 993 ± 5.7 and 811 ± 12.6 MPa, respectively, compared to the SHT and HIP + STA samples. In addition, the ultimate tensile strength and elongation values of the HIP + DA Inconel 718 superalloy were better than those of the alloys treated with SHT and HIP + STA.The dominant strengthening phase was the γ″ precipitate in the Inconel 718 superalloy, because the lattice misfits (ε) of γ/γ″ in all conditions possessed higher values than that of γ/γ′.The strengthening mechanisms of the Inconel 718 superalloy after SHT and HIP + STA treatments can be explained by the coherency strain strengthening mechanism due to the formation of fine γ″ precipitates. However, a combination of coherency strain strengthening and a dislocation-cut ordered particle strengthening mechanism is considered to be the reason for the strengthening of the Inconel 718 superalloy treated with HIP + DA.

## Figures and Tables

**Figure 1 materials-13-00151-f001:**
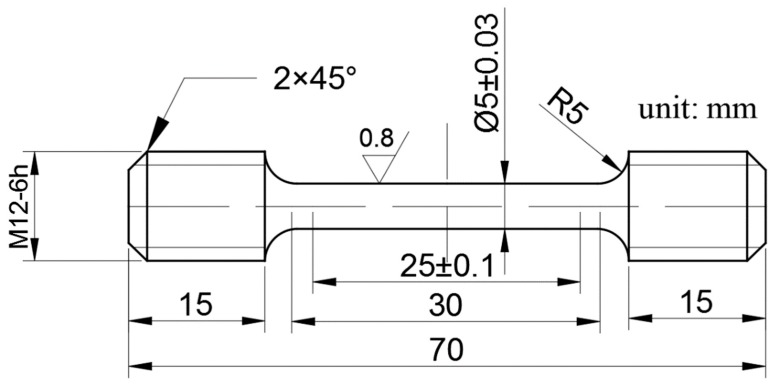
Dimensions of standard tensile specimens.

**Figure 2 materials-13-00151-f002:**
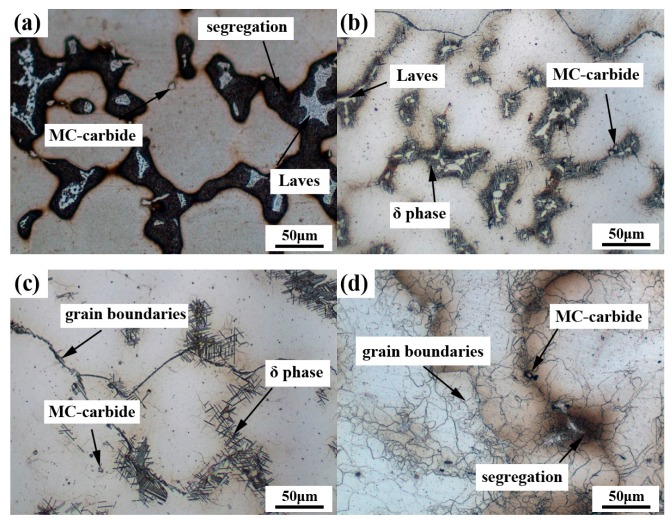
The optical microstructures of the Inconel 718 superalloy after different heat treatments: (**a**) the cast case; (**b**) standard heat treatment (SHT); (**c**) hot isostatic pressing + solution treatment + aging (HIP + STA); (**d**) HIP + direct aging (DA).

**Figure 3 materials-13-00151-f003:**
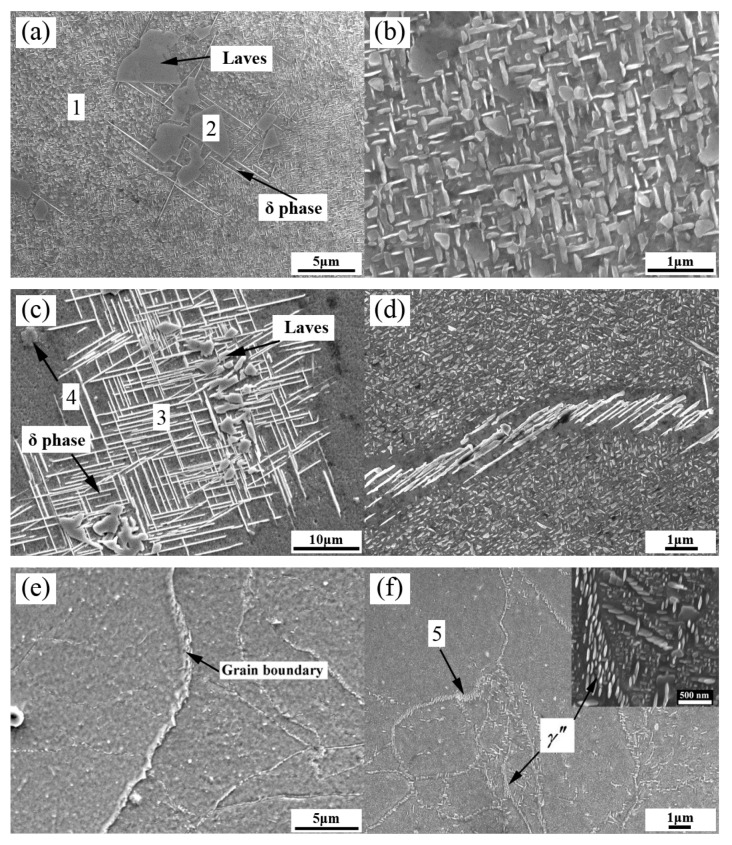
SEM images of the Inconel 718 superalloy after different heat treatments: (**a**,**b**) SHT; (**c**,**d**) HIP + STA; (**e**,**f**) HIP + DA.

**Figure 4 materials-13-00151-f004:**
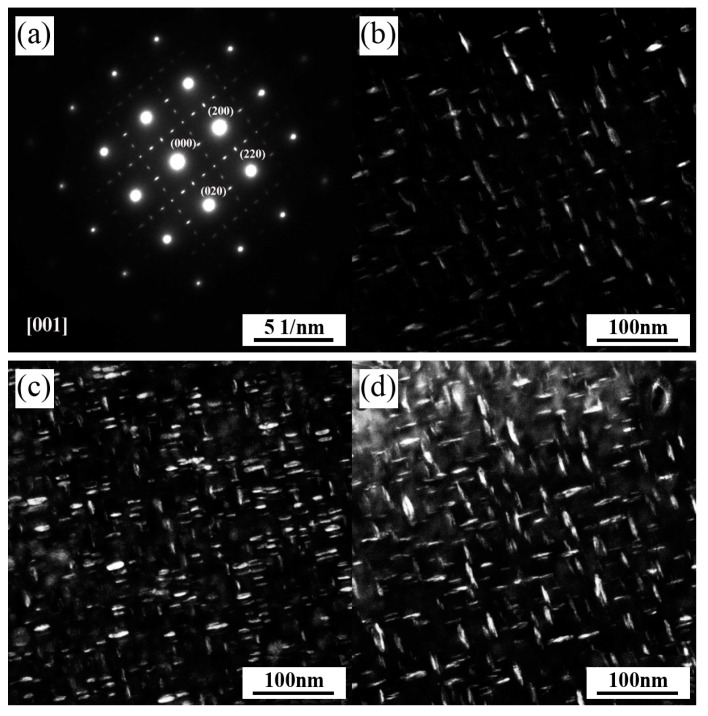
TEM images of the Inconel 718 superalloys showing γ″ phase: (**a**) selected area electron diffraction (SAED) pattern, dark field images for sample (**b**) SHT, (**c**) HIP + STA, and (**d**) HIP + DA.

**Figure 5 materials-13-00151-f005:**
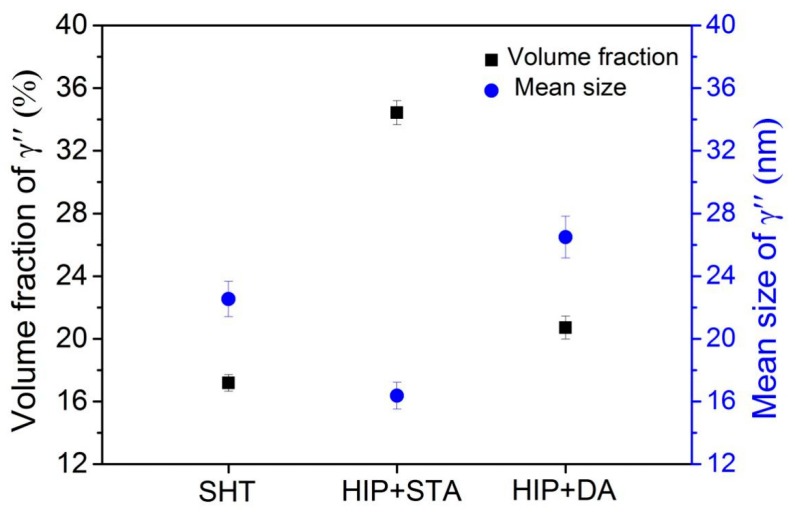
Volume fractions and mean sizes of γ″ precipitates in the SHT, HIP + STA, and HIP + DA Inconel 718 superalloys.

**Figure 6 materials-13-00151-f006:**
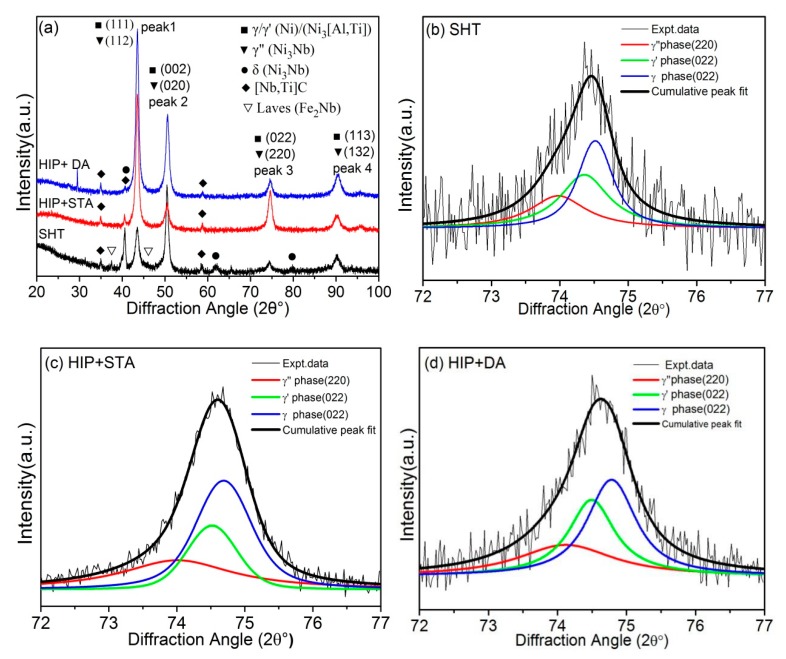
X-ray diffraction analysis of the SHT, HIP + STA, and HIP + DA Inconel 718 superalloys: (**a**) full XRD pattern; (**b**–**d**) deconvolutions of the diffraction peak 3 of the SHT, HIP + STA, and HIP + DA Inconel 718 superalloys, respectively.

**Figure 7 materials-13-00151-f007:**
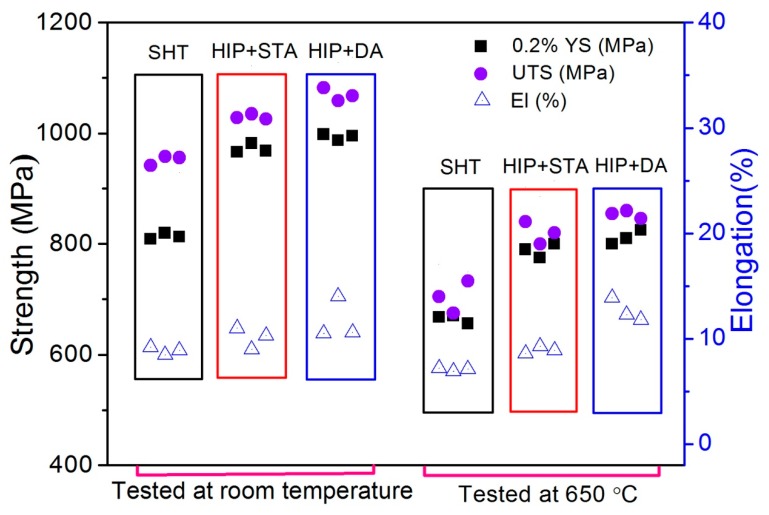
The tensile properties of the SHT, HIP + STA, and HIP + DA Inconel 718 superalloys at room temperature and at 650 °C.

**Figure 8 materials-13-00151-f008:**
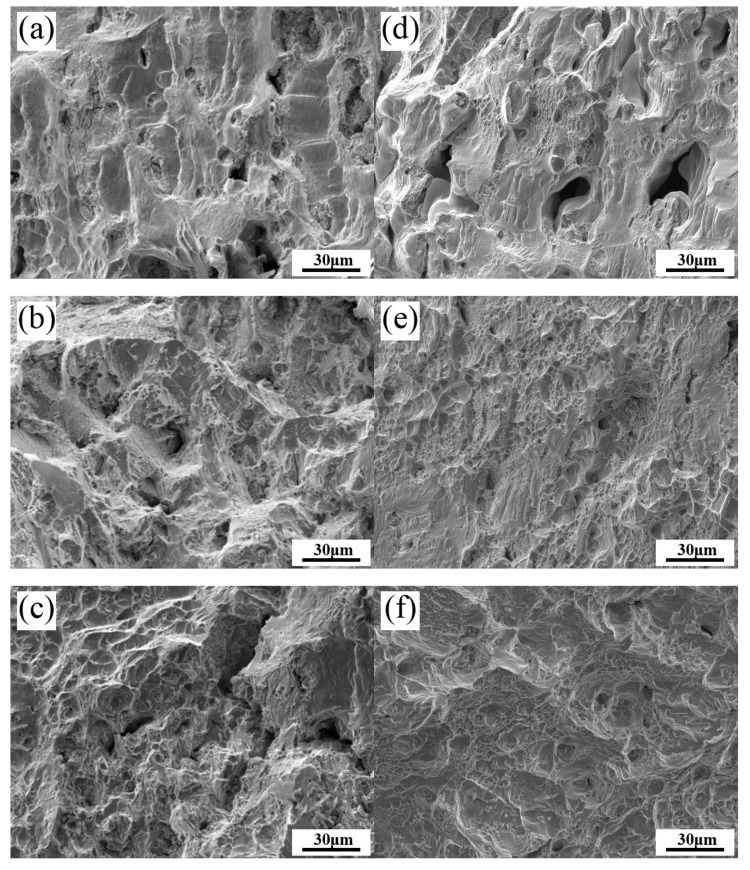
SEM fractographs of the tensile tested specimens of the Inconel 718 superalloys: SHT, HIP + STA, and HIP + DA at room temperature (**a**–**c**) and at 650 °C (**d**–**f**), respectively.

**Figure 9 materials-13-00151-f009:**
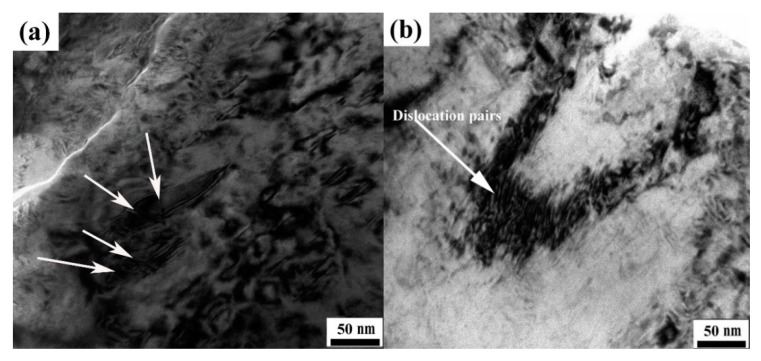
TEM micrographs of the deformed region of the tensile specimen after HIP + DA treatment: (**a**) relatively coarser γ″ precipitates and (**b**) dislocation pairs.

**Table 1 materials-13-00151-t001:** The results of the EDX analysis of the Inconel 718 superalloy after different heat treatments from Figure 3 (wt.%).

Spot	Phase Designation	Ni	Fe	Cr	Nb	Ti	C	Mo	Al
1	Matrix	52.48	18.99	19.68	2.47	0.86	2.61	2.46	0.35
2	Laves	37.39	11.25	12.11	29.71	1.07	–	8.09	0.35
3	δ	54.08	12.31	13.90	13.51	2.10	–	3.72	0.38
4	MC-carbide	2.61	1.12	0.88	66.81	7.51	13.69	7.31	0.38
5	γ″	53.21	11.28	12.80	13.61	2.12	–	6.62	0.36

**Table 2 materials-13-00151-t002:** Lattice parameters and misfits for the standard heat treatment (SHT), hot isostatic pressing + solution treatment + aging (HIP + STA), HIP + direct aging (DA) Inconel 718 superalloys.

Treatments	Lattice Parameter (nm)			Lattice Misfit ε (%)	
	γ Phase	γ′ Phase	γ″ Phase	γ/γ′	γ/γ″
SHT	a = 0.3601	a = 0.3607	a = 0.3620 c = 0.7408	0.1847	0.5407
HIP + STA	a = 0.3601	a = 0.3608	a = 0.3619 c = 0.7338	0.1869	0.4985
HIP + DA	a = 0.3603	a = 0.3610	a = 0.3626 c = 0.7492	0.1928	0.6320

**Table 3 materials-13-00151-t003:** The mean values of yield strength (YS), ultimate tensile strength (UTS), and elongation (EL) of the SHT, HIP + STA, and HIP + DA Inconel 718 superalloys at room temperature and at 650 °C.

Treatment	YS (MPa)	UTS (MPa)	EL (%)	YS(MPa)	UTS (MPa)	EL (%)
	Room Temperature			650 °C		
SHT	814 ± 5.6	952 ± 8.7	8.9 ± 0.4	666 ± 7.7	703 ± 29.1	7.1 ± 0.2
HIP + STA	972 ± 8.7	1030 ± 4.7	10.1 ± 1.0	788 ± 12.6	820 ± 20	8.9 ± 0.4
HIP + DA	993 ± 5.7	1070 ± 11.6	11.7 ± 2.0	811 ± 12.6	854 ± 10.0	12.7 ± 1.1
Wrought material	1035–1067	1275–1400	12–21	860–1000	1000–1200	12–19

**Table 4 materials-13-00151-t004:** The increase in yield strength ($\Delta \tau_{\gamma \prime \prime }$) of the SHT, HIP + STA, and HIP + DA Inconel 718 superalloys [12].

Treatments	Strengthening Method	SHT	HIP + STA	HIP + DA
$\Delta \tau_{1}$ (MPa)	coherent strain strengthening	232.83	337.04	316.43
$\Delta \tau_{2}$ (MPa)	dislocation-cut ordered particle strengthening	–	–	43.19

SHT: standard heat treatment; HIP + STA: hot isostatic pressing + solution treatment + aging; HIP + DA: hot isostatic pressing + direct aging.

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
