# Peer review of "Effects of the γ″-Ni3Nb Phase on Mechanical Properties of Inconel 718 Superalloys with Different Heat Treatments"

_materials, 2019, doi:10.3390/ma13010151_

Round 1
Reviewer 1 Report
The reviewed manuscript focuses on the strengthening mechanisms of the Inconel 718 superalloy after relative treatment, important issues in material engineering, therefore will be interesting for readers. A very interesting research methodology was used in the work and the obtained results are very interesting and well presented. The work is undoubtedly very valuable and very good therefore, I have only few remark/comment.
Title of paragraph 2.2 doesn’t fit its content, my suggestion: “Methodology of metallographic investigation”.
Fig 5. shows an unexpectedly high volume of γ" phase for HIP+STA, considering finesse precipitates it could be the effect of the binarization. In not high enough resolution the precipitation boundaries may have been counted to measured objects. Parameters of image analysis and image quality should be mentioned.
Fig. 7. YS for SHT at 650oC seems to be higher than UTS. Is it possible? How has it been measured?
Reviewer 2 Report
NO COMMENTS
Reviewer 3 Report
The authors talk about an important alloy, Inconel 718, that has significant industrial values. Even, this alloys has been an ideal material for studying the effect of novel processing methods such as HIP, 3D printing etc.
The authors have put a good dataset of grain boundary strengthening imparted by gamma-double prime.
This paper should be published, however I have a comment on Conclusion 3. The normal X-ray source is not powerful enough to isolate the peaks of three phases. Usually, small angle x-ray scattering such as Synchrotron is used to obtain better values of lattice mismatch. Hence, the authors should discuss of the limitation of their analysis using normal x-ray diffraction.
Reviewer 4 Report
The topic presented in the manuscript “Effects of the γ″-Ni3Nb phase on Mechanical Properties of Inconel 718 superalloys with different heat treatments” is very interesting. The manuscript is well written and presents novel results. The manuscript makes clear the potential of the use of HIP treatment to replace the standard heat treatment, due to the improvement of mechanical properties. Here it would have been interesting to know the exact cooling rate used after the HIP cycle, due to the great effect it has on the final microstructures and properties. In addition, a justification is needed about why the used temperature in the HIP (1170°C) is higher than that used in the standard heat treatment (1095°C) and why 4h are selected for the HIP treatment.
In general, it is difficult to draw conclusions about the real effect of applying hot isostatic pressure on mechanical properties, when the other parameters, temperature and holding time also differ for the standard heat treatment conditions.
Please improve the captions in Figure 4 and Figure 8 to make them clearer.
It is not clear whether Figures 9a and 9b correspond to the same region or no.
Reviewer 5 Report
This work concerns the study of microstructure and mechanical properties evolution of Inconel 718 subjected to different post-processing.
The work presents interesting points.
However, some parts need further details in order to indicate the innovative with respect to the current literature.
Therefore, I would recommend reconsidering the manuscript for publication after major revision.
Please find below the aspects that require modifications:
Introduction
Line 42-44: there are other post-processing to homogenize the microstructure (nor only HIP). Explaining HIP treatments, it should also be reported its ability to close pores as well as a possible limitation regarding cooling rates and the cost Please reports clearly the innovative with respect to the current literature on the gamma double prime evolution. Line 66-68: the authors wrote, “rarely publications”. I suggest adding a few references to this sentence. There are studies on gamma double prime for Inconel 718. Therefore, the innovative finding should be clearly reported.
Experimental procedure
How was evaluated the chemical composition? How many tensile samples were tested for each condition?
Results
(3.1 Microstructure characterization): The microstructure should be compared to the IN718 processed in similar conditions revealing similarly and deviations. Were performed EDS analyses on the samples? (3.3 Mechanical properties): The authors should compare the mechanical properties with Inconel 718 processed in similar conditions in order to show the results in the current literature.
Round 2
Reviewer 2 Report
The authors do not want to understand that it is necessary to associate semi-quantitative investigations (EDX) to SEM images for a preliminary identification of particular phases or precipitates in microstructural images, as instead indicated in the paper. XRD spectra certify stoichiometrically the phase or compound whose structural development is further certified by TEM. Without reporting the local chemical composition with EDX on precipitation images, the reader does not distinguish between the different precipitations.Author Response
Please see the attachment.

Reviewer 5 Report
The authors made a limited amount of modifications with respect to the previous version.
A more significant effort is required in order to improve the quality of the manuscript, making it suitable for publication.
The microstructure required further explanations, compared with the literature.
For instance, (Fig.2):
The as-cast state (Fig. 2a) exhibits MC carbides. These carbides seem not present for the other conditions while they could form during the air cooling. Considering the dissolution temperature of carbides as well as the presence of air cooling after the heat treatments, the formation of the carbides should occur.
HIP+DA seems to have a finer microstructure of HIP+SHT and SHT conditions. Probably it derives from an effect of the precipitation along the grain boundaries. However, it should explain why this is not evident for the HIP+SHT condition.
Therefore, the resulted phases should be explained taking into account the TTT diagram of the alloy, also considering other papers.
For the mechanical properties:
There are published papers with the mechanical properties of heat-treated or HIPed Inconel 718 alloy. The data could be compared with some of these work indicating possible similarities.
Please report the standard deviation of the mechanical properties (Table 2).
Additionally, standard values are reported for heat-treated IN718, and therefore, a discussion may improve the quality of the paper.
For instance:
doi.org/10.1016/S0921-5093(03)00079-0
Finally:
The chemical composition method used for evaluating the chemical composition should be reported in the paper.
